# Mycelium-mediated transfer of water and nutrients stimulates bacterial activity in dry and oligotrophic environments

Anja Worrich[1,2,†], Hryhoriy Stryhanyuk[3], Niculina Musat[3], Sara König[2,4], Thomas Banitz[4], Florian Centler[2], Karin Frank[4,5,6], Martin Thullner[2], Hauke Harms[2,5], Hans-Hermann Richnow[3], Anja Miltner[1], Matthias Kästner[1,*] & Lukas Y. Wick[2,*]

Fungal–bacterial interactions are highly diverse and contribute to many ecosystem processes. Their emergence under common environmental stress scenarios however, remains elusive. Here we use a synthetic microbial ecosystem based on the germination of *Bacillus subtilis* spores to examine whether fungal and fungal-like (oomycete) mycelia reduce bacterial water and nutrient stress in an otherwise dry and nutrient-poor microhabitat. We find that the presence of mycelia enables the germination and subsequent growth of bacterial spores near the hyphae. Using a combination of time of flight- and nanoscale secondary ion mass spectrometry (ToF- and nanoSIMS) coupled with stable isotope labelling, we link spore germination to hyphal transfer of water, carbon and nitrogen. Our study provides direct experimental evidence for the stimulation of bacterial activity by mycelial supply of scarce resources in dry and nutrient-free environments. We propose that mycelia may stimulate bacterial activity and thus contribute to sustaining ecosystem functioning in stressed habitats.

[1] Department of Environmental Biotechnology, UFZ—Helmholtz Centre for Environmental Research, Permoserstraße 15, 04318 Leipzig, Germany. [2] Department of Environmental Microbiology, UFZ - Helmholtz Centre for Environmental Research, Permoserstraße 15, 04318 Leipzig, Germany. [3] Department of Isotope Biogeochemistry, UFZ—Helmholtz Centre for Environmental Research, Permoserstraße 15, 04318 Leipzig, Germany. [4] Department of Ecological Modelling, UFZ—Helmholtz Centre for Environmental Research, Permoserstraße 15, 04318 Leipzig, Germany. [5] German Centre for Integrative Biodiversity Research (iDiv) Halle-Jena-Leipzig, Deutscher Platz 5e, D-04103 Leipzig, Germany. [6] University of Osnabrück, Institute for Environmental Systems Research, Barbarastaße 12, 49076 Osnabrück, Germany. † Present address: German Centre for Integrative Biodiversity Research (iDiv) Halle-Jena-Leipzig, Deutscher Platz 5e, 04103 Leipzig, Germany. * These authors contributed equally to this work. Correspondence and requests for materials should be addressed to N.M. (email: niculina.musat@ufz.de).

Fungi and bacteria co-inhabit a wide variety of environments[1,2] and their interactions are significant drivers of important ecosystem functions and services[3]. In nature, they live in habitats exposed to fluctuating environmental conditions and frequently underlie strong selection pressure by limiting resources or drought[4]. To survive periods of stress, bacteria enter reversible states of low metabolic activity or dormancy[5]. Dormancy transiently disables bacterial functions while generating a seed bank of resting bacteria with the potential to be resuscitated in response to favourable environmental changes[5,6]. Fungi, by contrast, have been shown to possess higher resistance to drought and nutrient limitation due to efficient resource translocation between spatially separated sources and sink regions in their mycelia[7–10]. The redistribution of water in fungal mycelia was shown to enhance ecosystem resistance by preserving fungal carbon mineralization during drought[11]. While recent data suggest that mycelia create hospitable microhabitats for bacteria due to the exudation of carbonaceous compounds[2,12,13] and a moistening of the surrounding substrate[11], experimental evidence for a stimulation of bacterial activity in harsh environments is lacking.

During the past decade, powerful methods have been developed for the investigation of microbial interactions down to the single-cell scale[14,15]. Numerous studies revealed that resource exchange is a successful strategy in microbial communities to cope with unfavourable environmental conditions[16–19]. However, the role of resource transfer processes from fungi to bacteria for the maintenance of bacterial activity in dry and oligotrophic habitats remains to be elucidated.

In the present study, we use a synthetic microbial ecosystem to examine whether mycelia of fungi and oomycetes reduce water and nutrient stress for bacteria and thus enable bacterial activity in otherwise dry and oligotrophic environments. We introduce initially inactive spores of the soil-bacterium *Bacillus subtilis* into a water- and nutrient-free area and several hyphal model organisms, growing on a physically separated water and nutrient-rich agar piece, were allowed to overgrow the spore-bearing region. We find that the presence of mycelia enables both the germination of bacterial spores as well as vegetative growth near the hyphae. Spatially resolved secondary ion mass spectrometry in combination with stable isotope labelling reveals a long distance transport of labelled compounds in mycelia and a supply of water, carbon and nitrogen to the cells of *B. subtilis* located close to the hyphae. Spores more distant from the mycelium remain dormant and show no enrichment of the labelled substrates. Our results demonstrate that mycelium-forming fungi and oomycetes facilitate bacterial activity in dry and oligotrophic environments by providing nitrogen, carbon and water to bacteria and thus improving their habitat conditions.

## Results

**Mycelia affect spore germination and growth of *B. subtilis*.** We developed a synthetic microbial ecosystem to assess whether fungal–bacterial interactions emerge in presence of drought and nutrient limitation stress conditions. Specifically, we tested to what degree the presence of mycelia evokes beneficial shifts in the habitat conditions for bacterial activity and growth. To this end, we placed spores of *B. subtilis* on silicon wafers (Fig. 1a) and analysed their germination and growth as an indicator for access to water and nutrients. The ascomycete *Fusarium oxysporum*, the basidiomycete *Lyophyllum* sp. Karsten, and the mycelium-forming oomycete *Pythium ultimum* were inoculated to a water and nutrient-rich agar patch physically separated from the spore-bearing silicon wafer (Fig. 1a). The mycelium overgrew the wafer within two (*P. ultimum*), four (*F. oxysporum*) and five days

(*Lyophyllum* sp. Karsten) in direction of a second nutrient and water reservoir at the opposite side of the wafer. Subsequently, all biomass was detached from the wafer and the abundance of *B. subtilis* cells in presence and absence of mycelia determined by counting total colony forming units (c.f.u.; Fig. 2a). The number of total c.f.u. detached from mycelia-free controls $((1.0 \pm 0.5) \cdot 10^5$ c.f.u.) corresponded to the number of spores applied $((1.0 \pm 0.3) \cdot 10^5$ c.f.u.) and thus excluded harmful wafer effects on spore germination (Fig. 2b). When hyphae overgrew the spore region, however, the total average c.f.u. numbers increased (Fig. 2b), indicating that mycelia supported growth of bacteria in dry and nutrient-free regions on the wafer. The highest c.f.u. numbers of *B. subtilis* were observed in presence of the oomycete *P. ultimum* $((3.5 \pm 1.8) \cdot 10^5$ c.f.u.), followed by *Lyophyllum* sp. Karsten $((2.8 \pm 0.6) \cdot 10^5$ c.f.u.) and *F. oxysporum* $((2.7 \pm 0.7) \cdot 10^5$ c.f.u.). Furthermore, we analysed the number of spores to obtain information on the fraction of sporulated and germinated cells relative to total cell numbers (Fig. 2a). In the control, we found that $100 \pm 6\%$ of the total c.f.u. remained in form of spores on the wafers. However, on *P. ultimum* wafers only $10 \pm 2\%$ of the total cells were dormant (Fig. 2c). For *F. oxysporum* and *Lyophyllum* sp. Karsten the fraction of spores was higher ($34 \pm 3\%$ and $24 \pm 3\%$, respectively), but vegetative cells still accounted for the majority of the total c.f.u. (Fig. 2c). Microscopic analyses revealed the presence of vegetative *B. subtilis* cells always in close vicinity to the hyphae of *P. ultimum* (Fig. 1b,c magenta arrows). At locations more distant to hyphae however, *B. subtilis* persisted in the spore form (Fig. 1b,c yellow arrows).

**Chemical mapping of *B. subtilis* and *P. ultimum* with ToF–SIMS.** ToF–SIMS (time of flight- secondary ion mass spectrometry) analyses without prior isotope labelling were used to map the structural arrangement and elemental composition of the mycelium, vegetative bacterial cells and bacterial spores directly on the silicon wafer without previous isolation. This method prevented us from losing spatial and chemical information typically occurring during sample preparation in SIMS experiments[20]. The mass peaks corresponding to $O^-$, $OH^-$, $PO_2^-$, $CH^-$, $CN^-$ and $S^-$ secondary ions were selected to represent the variance in composition of the biomass and its extracellular environment. To reduce the intensity modulation due to topography and effects related to the density of the sample, we normalized the intensity in respective ion yield distribution maps to total ion counts. As in microscopy, the ToF–SIMS analysis showed that vegetative cells of *B. subtilis* were located in close vicinity to the hyphae, whereas the spores of *B. subtilis* were randomly distributed regardless of the presence of hyphae (Fig. 3). The *B. subtilis* spores, which are known to be rich in sulfur compounds[21], were observed as bright spots in the normalized intensity maps of $O^-$ and $S^-$ species (Fig. 3a,f) and vegetative cells were detected by their enhanced intensity of $OH^-$ ions (Fig. 3b), respectively. The $PO_2^-$ and $CH^-$ ion signal intensities however, could not be used for detailed biomass analysis, as a diffuse, strong $PO_2^-$ (Fig. 3c) and $CH^-$ (Fig. 3d) ion yield, distributed over the whole field of analysis, precluded clear discrimination between hyphae, spores and vegetative cells and pointed at the presence of extracellular material, possibly due to excretion, lysis of hyphae or damaged bacterial cells. By contrast, high secondary ion yields for $CN^-$ (Fig. 3e) reflected locally distinct protein contents and likely revealed locations of vegetative bacterial cells (white arrows) and residual hyphal fragments.

**NanoSIMS analysis of water and nutrient transfer.** Building on the ToF–SIMS chemical mapping, we used nanoSIMS and isotope

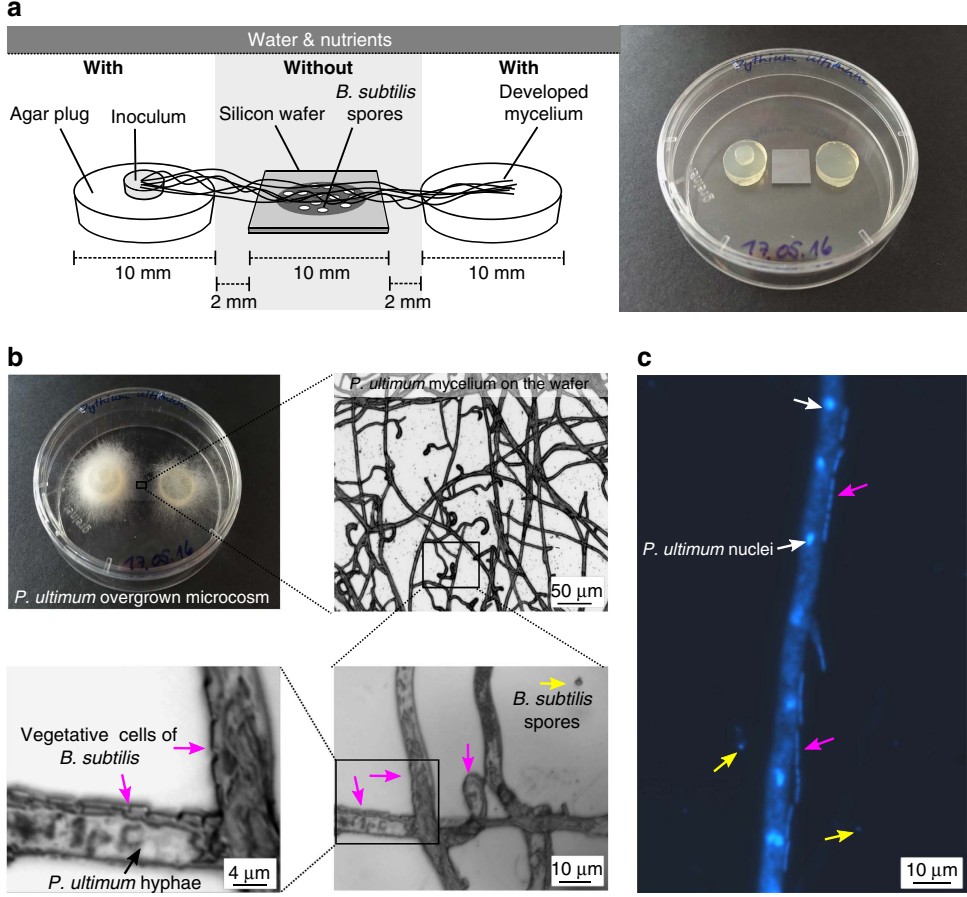

**Figure 1 | Synthetic microbial ecosystem reveals spore germination in presence of mycelia in dry and oligotrophic environments.** (**a**) Scheme and photographs of the setup employed to carry out the germination, growth and labelling experiments. The synthetic ecosystem is comprised of two agar plugs serving as water and nutrient sources ('with') for the fungi or the oomycete inoculated on top of one of the agar plugs. A silicon wafer free of water and nutrients ('without') placed in the middle between the two 'with' zones served as carrier for spores of *B. subtilis*. An air gap between the 'with' and 'without' zone prevented the diffusion of water or substrates to the spore region. (**b**) Gradual enlargement of bright-field micrographs of the silicon wafer overgrown by mycelium of *P. ultimum*. In close vicinity to the hyphae (black arrow) rod-shaped, vegetative bacterial cells (magenta arrows) were found, whereas smaller, round-shaped spores (yellow arrows) were located more distantly. (**c**) Fluorescence micrograph of the 4′,6-diamidine-2-phenylindol (DAPI)-stained wafer showing *P. ultimum* hyphae containing nuclei (white arrows) and vegetative cells as well as spores of *B. subtilis*.

labelling approaches to obtain quantitative information on the nutrient and water transfer from the hyphae of *P. ultimum* to spores and vegetative cells of *B. subtilis*, respectively. The distribution of $^{18}O$, $^{13}C$ and $^{15}N$ in hyphae, vegetative cells and spores was quantified after the mycelium of *P. ultimum* had overgrown the spore-bearing wafer (Figs 4 and 5). *P. ultimum* was grown in parallel experiments with $^{18}O$-labelled water or with a combination of $^{13}C$-glucose and $^{15}N$-ammonium sulfate, to prevent slow development of hyphae due to strong kinetic isotope effects[22,23]. The localization of the biomass and its integrity was assessed by $^{12}C^{14}N^{-}$ detection in two different fields of analysis (Fig. 4a,c). Both, vegetative cells and spores consistently provided strong $^{12}C^{14}N^{-}$ signals, whereas hyphae of *P. ultimum* were found to be intact only in one of the two observation fields studied (Fig. 4a,c). As for microscopy and ToF–SIMS, vegetative cells of *B. subtilis* were either observed in close vicinity to the intact hyphae or lysed hyphal biomass. Strikingly, vegetative cells were consistently positioned longitudinally along the hyphae, forming a regular stratification typical for *B. subtilis* biofilms[24] (Fig. 4c). Due to entire $^{12}C$ and $^{14}N$ replacement by their corresponding stable isotopes neither the hyphae nor the hyphae-attached vegetative cells were detected by the $^{12}C^{14}N^{-}$ signal in $^{13}C$ and $^{15}N$ labelling experiments. High $^{12}C^{14}N^{-}$ signals,

however, were seen in spores distant from the hyphae as was further confirmed by the secondary electron image (Fig. 5a).

Fungal biomass was found to be uniformly enriched in $^{18}O$ except for small holes probably arising from ongoing hyphal disintegration (Fig. 4b). Vegetative cells in close vicinity to the hyphae showed higher enrichment in $^{18}O$ ($2.8 - 8.9\%$) than spores located at longer distance to hyphae ($1. - 3.1\%$; Fig. 6a) suggesting $^{18}O$ transfer from the hyphae to vegetative cells. Compared with intact hyphae ($12.1 \pm 0.0\%$ excess over natural abundance), $^{18}O$ enrichment of *B. subtilis* cells however, was lower. This is presumably due to the much smaller diameter and the dilution of the signal attributed to the spores' own oxygen-containing molecules or the core water (which can account for up to 27–55% of the core wet weight[25]). Enrichment levels of $^{18}O$ in spores was slightly higher in the $^{18}O$ labelling experiments compared with non-labelled samples (Fig. 6a; Supplementary Fig. 1). Thus, some transport of water must have occurred via the gas phase. The transfer of carbon and nitrogen containing nutrients from hyphae to the bacterial cells was underpinned by significant $^{13}C$ and $^{15}N$ enrichment in hyphae and vegetative cells vicinal to hyphae yet not in spores more distant to hyphae (Fig. 5b,c). Relative to the $^{15}N$ and $^{13}C$ enrichment of the hyphae ($^{15}N$: $37.2 \pm 0.0\%$ and $^{13}C$: $34.6 \pm 0.0\%$), vegetative bacterial

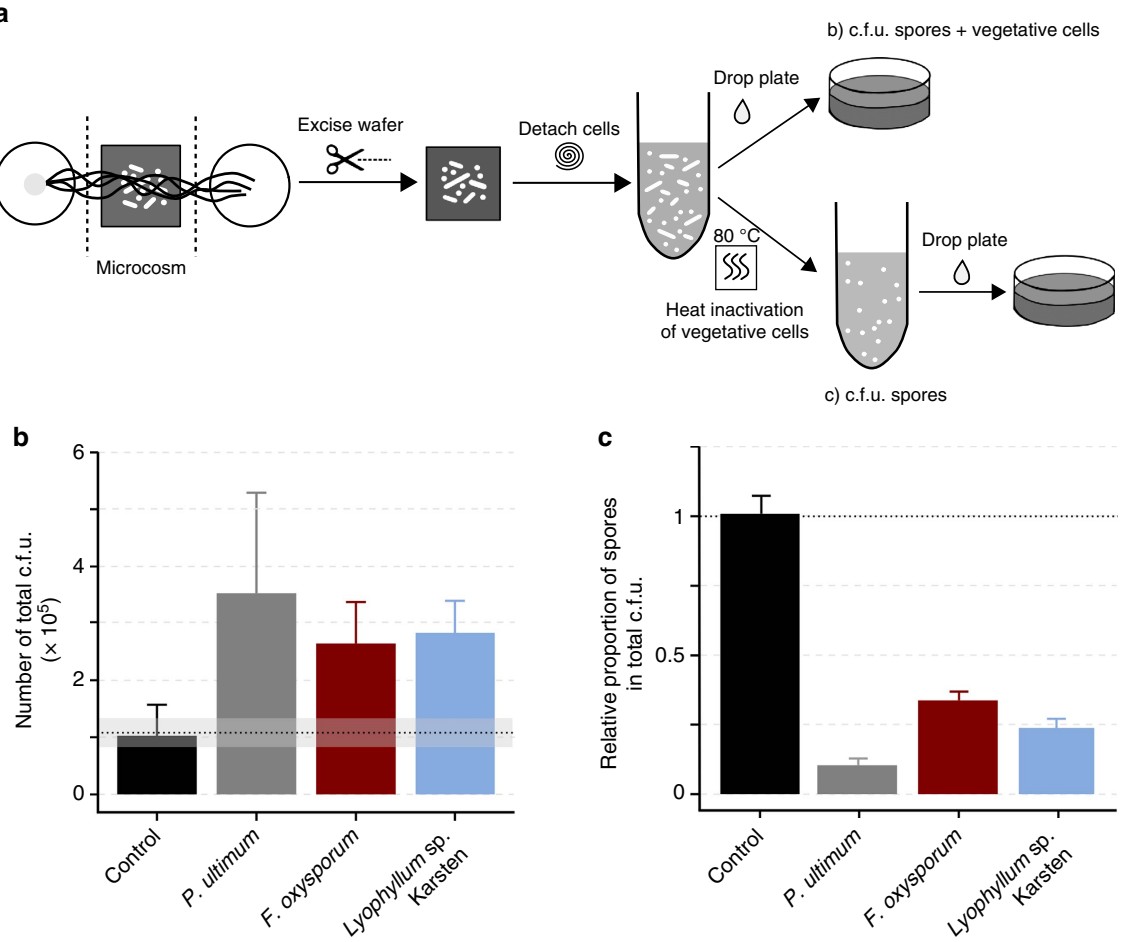

**Figure 2 | The presence of mycelium enables bacterial growth and spore germination. (a)** Scheme of the experimental procedure providing information about vegetative growth and germination in presence of mycelium. Total cell number was determined as c.f.u. after cell detachment from the wafer and plating on agar. Spores were obtained by counting c.f.u. after heat-inactivation of the vegetative cells. **(b)** c.f.u. of *B. subtilis* after detachment from the control wafer (no contact to mycelium) and wafers overgrown by *P. ultimum*, *F. oxysporum* or *Lyophyllum* sp. Karsten. The dashed line shows the number of c.f.u. applied to the wafer with the inoculum. In presence of mycelium, the number of total c.f.u. increased compared with the control. Bars show the average number of c.f.u. and error bars indicate the s.d. **(c)** Respective proportions of *B. subtilis* spores, calculated by dividing the number of spore c.f.u. by the total cell c.f.u. determined in **b**. The bars show the mean of the quotients for the three replicates and the s.d. The number of spores was different from the number of total cells for the respective strain but did not differ for the control.

cells were more enriched in $^{15}N$ $(38.3-70.1\%)$ than in $^{13}C$ $(20.1-47.1\%)$ indicating both a lower dilution by spore-inherent nitrogen compounds and a higher nitrogen demand by germinating spores and growing cells of *B. subtilis*, respectively. By contrast, spores were $^{13}C$ and $^{15}N$ enriched by $1.2-4.8\%$ and $0.6-5.1\%$, respectively, and showed C and N isotope abundances comparable to non-labelled samples (Fig. 6b,c; Supplementary Fig. 2). The $^{18}O$ and the joint $^{13}C$ and $^{15}N$ enrichments, hence, provided clear evidence of transfer of water and nutrients from hyphae to bacterial cells and the importance of the relative spatial organization of spores and hyphae on the wafer surface, respectively. The low levels of stable isotope incorporation into bacterial spores distant to the hyphae and the observation that vegetative cells were solely found in close proximity to hyphae indicate a very low air-borne transfer of the isotope labels (for example, by humidity or fungal volatiles) to the spores.

## Discussion

By verifying hyphal uptake, translocation and transfer of $^{13}C$, $^{15}N$ and $^{18}O$ labels to subsequently germinating *Bacillus subtilis*

spores, our results demonstrate that mycelia enable bacterial activity in regions devoid of water and nutrients. Utilizing a synthetic microbial ecosystem and a novel combination of ToF- and nanoSIMS quantitative imaging of single-cell trophic interactions we showed that (i) the presence of fungi and oomycetes in an initially dry and nutrient-free environment induced the germination of metabolically inactive spores of *B. subtilis* to actively growing cells, (ii) only spores in close vicinity to the hyphae could germinate, and (iii) a transfer of nutrients and water from hyphae to the bacterial cells underlay this activation. Thus, our study provides for the first time direct experimental evidence for stimulation of bacterial activity by mycelial supply of scarce resources in dry and nutrient-free environments.

Within microbial communities competitive and cooperative interactions are often related to the excretion and uptake of metabolites from involved partners[17,26,27]. It is widely assumed that fungal exudates are a source of nutrients for bacteria in the mycosphere including organic acids[28], sugars, polyols[29] and amino acids[29]. In addition to known germinants such as amino acids and sugars[30], the availability of water is required for spore germination. Water uptake by the spore enables the hydration of

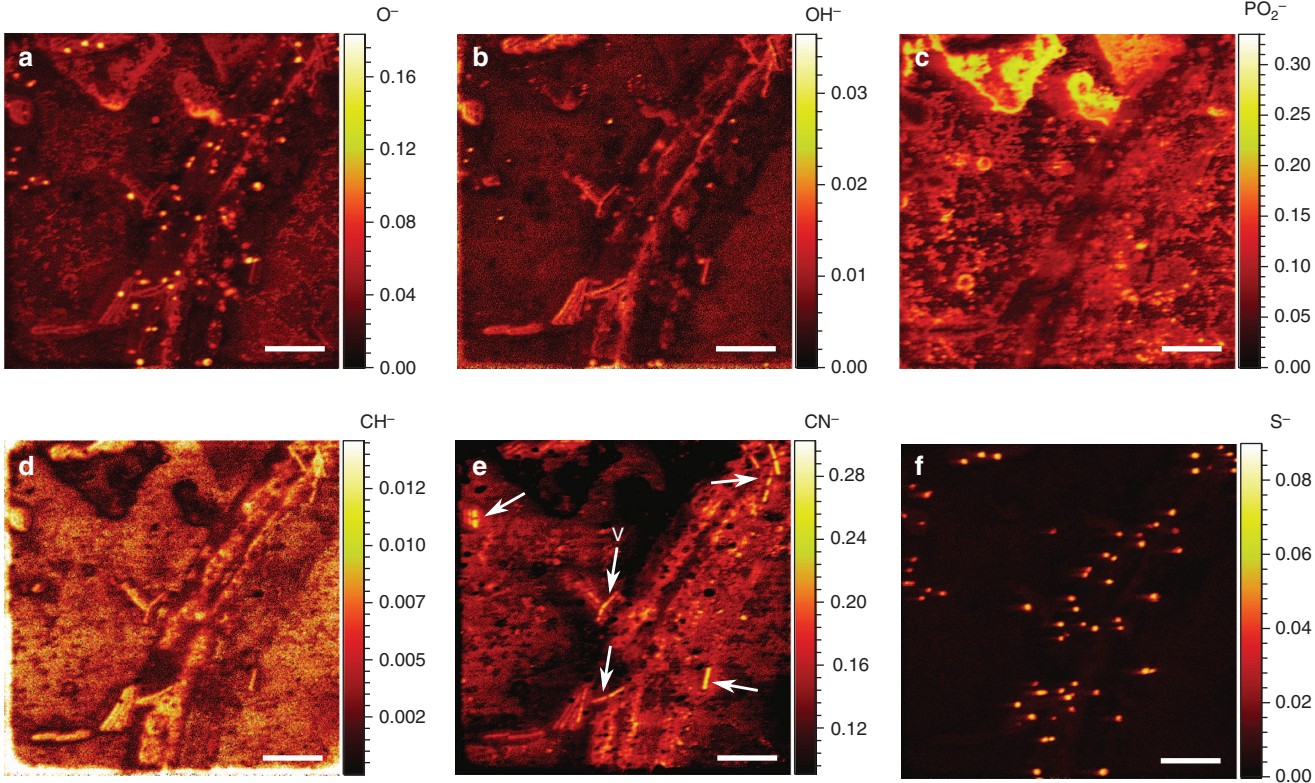

**Figure 3 | ToF–SIMS reveals sample composition via yield of secondary ion species.** Mass-resolved chemical map of *P. ultimum* hyphae as well as vegetative cells and spores of *B. subtilis* on top of the silicon wafer. Six negative secondary ion species were detected (**a**–**f**) in a 56 × 56 µm sample area. ToF–SIMS images produced were normalized by the total ion counts. The colour scale indicates the relative ion counts for each secondary ion species with warmer colours representing higher relative ion counts and cooler colours representing lower relative ion counts. Arrows in **e** point to vegetative cells (V). Scale bars, 10 µm.

the spore core and represents a crucial step in the germination process[30]. Recently, fungal hyphae were shown to redistribute water and thereby compensate differences in the soil matric potential[11]. Although the authors provided evidence that such water redistribution drastically increased the functional ecosystemic stability under drought, the consequences of this underrated pathway for other soil organisms remained unexplored. Our results suggest that hyphal water translocation may not only provide fungi with a higher resistance to drought, but also facilitates bacterial activity in close proximity to the hyphae. Especially at low soil water potentials, the mycosphere may thus form a hotspot of microbial activity and significantly foster the maintenance of important ecosystem functions and services.

Three different strategies are described by which bacteria may derive nutrition from fungi: necrotrophy, extracellular biotrophy and endocellular biotrophy. In necrotrophic interactions, bacteria secrete molecules that kill the fungus and thus induce a release of nutrients. Extracellular biotrophs live in close proximity and use nutrients exuded from the hyphae, whereas endocellular biotrophs live inside the fungus and take up nutrients from the cytoplasm[31]. In accordance with existing literature, we never observed any endocellular occurance of *B. subtilis*. Hence, we consider endocellular biotrophy to be rather unlikely. Indeed, vegetative cells were always located tightly arranged in a longitudinal orientation along the hyphae resulting in maximum contact surface between the bacterial cell and the hyphae. This strongly indicates extracellular biotrophy based on the consumption of fungal exudates[31]. The regular cell chains observed along the hyphae (Fig. 5a) most probably arose from

subsequent growth of the cells leading to a colonization of the hyphae as usually observed for the interaction of *B. subtilis* with plant roots[32]. In agriculture *B. subtilis* strains are routinely applied to soil because their biocontrol activity was shown to protect plants against fungal infestation[33]. Active killing of fungal hyphae in necrotrophic interactions was described for different *Bacillus* strains[34,35] and most likely induced the disintegration of the hyphae observed in the course of our SIMS experiments. However, before germination the metabolically inactive spores cannot have evoked killing of the hyphae. Moreover, we observed vegetative cells along undamaged hyphae of *P. ultimum* still containing intact nuclei (Fig. 1c). This demonstrates that neither necrotrophy nor nutrients leaking from dead hyphal cells can be solely responsible for the stimulation of bacterial activity in the vicinity of mycelia. The dry conditions on the wafer strongly restricted the diffusion of exudates and thus only spores in close contact to the mycelium germinated. Indeed, already at distances of more than 1.7 µm spores did not germinate (cf. Fig. 5a). Thus, the close proximity between the mycelium and the bacterial cells constituted a prerequisite for diffusion-based transfer of resources as observed in different studies on nutrient exchange[36] or chemical communication[37] in microbial communities. However, under more humid conditions in soil, exudate diffusion probably extends beyond the direct vicinity of the hyphae.

Our observation that fungi support the growth of bacteria is in accordance with studies that showed an increased number of culturable bacteria in the vicinity of different fungal hyphae in soil microcosms[12]. Given the evidence that bacteria are attracted to the mycosphere[38], it is tempting to speculate that the local conditions in these regions are often favourable for bacterial

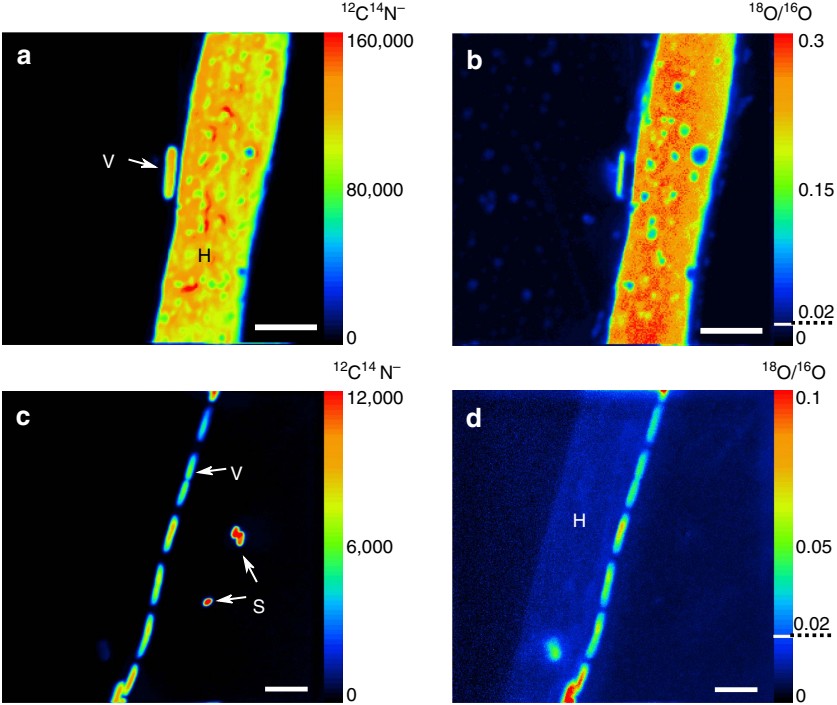

**Figure 4 | Water is transferred from hyphae to bacterial cells in dependence of the spatial organization.** NanoSIMS images of *P. ultimum* hyphae (H), *B. subtilis* spores (S) and vegetative cells (V) identified in the total biomass ($^{12}C^{14}N^-$) images (**a,c**) of $^{18}O$-enriched samples. The ratio images of $^{18}O/^{16}O$ show the incorporation of label from water from the nutrient-rich zones into the biomass of *P. ultimum* and *B. subtilis* (**b,d**). The colour scale indicates the intensities of $^{12}C^{14}N^-$ (**a,c**) and enrichment in $^{18}O$ (**b,d**) with warmer colours representing higher secondary ion counts (**a,c**) or enrichment (**b,d**) levels and cooler colours representing lower values. Dashed lines indicate natural abundance of $^{18}O$. Images represent different fields of analysis corresponding to sample areas of $20 \times 20$ (**a,b**) and $30 \times 30\,\mu m$ (**c,d**). Scale bars, 4 μm.

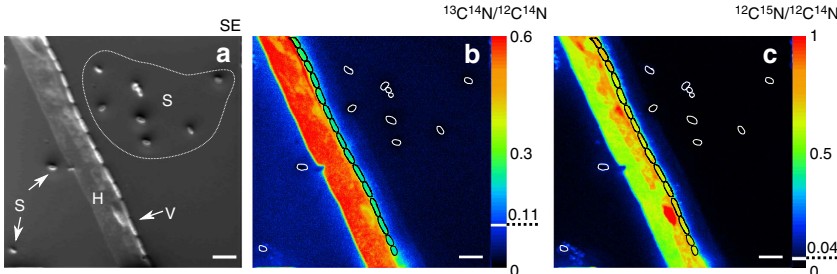

**Figure 5 | Bacterial cells in close proximity to the hyphae receive carbon and nitrogen.** NanoSIMS images of *P. ultimum* hyphae (H), *B. subtilis* spores (S) and vegetative cells (V) identified in the secondary electron (SE) image (**a**) of $^{13}C$ and $^{15}N$ -enriched samples. The ratio images of $^{13}C^{14}N/^{12}C^{14}N$ (**b**) and $^{12}C^{15}N/^{12}C^{14}N$ (**c**) show the incorporation of label into the biomass of *P. ultimum* and *B. subtilis*. The colour scale indicates the relative enrichment in $^{13}C$ and $^{15}N$ (**b,c**) with warmer colours representing higher enrichment levels and cooler colours indicating lower values. The higher background signal for $^{13}C$ on the left side of the hyphae (**b**) results from the topography-dependent re-deposition of sputtered sample material. Dashed lines indicate natural abundances of $^{13}C$ and $^{15}N$. Vegetative cells and spores are framed in black and white, respectively. Images represent a field of analysis corresponding to a sample area of $40 \times 40\,\mu m$. Scale bars, 4 μm.

activity. Fungal mycelia were shown to change local habitat conditions and thereby affect bacteria by changing the water availability, the pH, the soil structure or by secreting inhibitory or stimulatory compounds[39]. Thereby they create, modify or destroy habitats and thus pursue 'ecosystem engineering', which may influence the ecological success of other species[40]. In our experiments, representatives of different fungal phyla as well as a mycelium-forming oomycete could induce the germination and growth of *B. subtilis* by the supply of sufficient amounts of water and nutrients. The differences observed for total c.f.u. and the amount of germinated cells may be explained by the differences in the mycelial coverage of the wafer or differing cell wall compositions of oomycetes and fungi such as the lack of chitin in

*P. ultimum*. Although the combination of the selected organisms may be artificial, we observed similar net effects for all three mycelial organisms likely indicating widespread nutrient and water transfer from mycelia to bacterial cells. The emergence of the interaction under the extremely harsh environmental conditions prevailing in the synthetic microcosms strongly suggests that the observed phenomenon could also arise under a wide range of less extreme conditions found in natural systems. The finding that mycelia facilitate bacterial activity by multiple resource supply may help to further understand the small-scale dynamics of bacterial activity in heterogeneous ecosystems exposed to fluctuating environmental conditions. The ability of mycelia to redistribute and transfer resources allows bacteria to

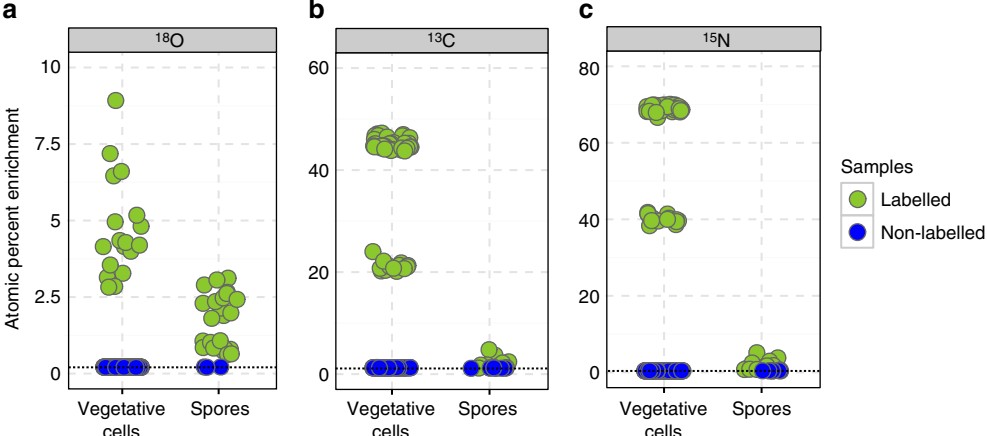

**Figure 6 | Vegetative cells are higher enriched in $^{13}$C, $^{15}$N and $^{18}$O compared with spores of *B. subtilis*.** Atom percent enrichment (APE) for (**a**) $^{18}$O, (**b**) $^{13}$C and (**c**) $^{15}$N by single cells and spores of *B. subtilis* measured with nanoSIMS in labelled (green) and non-labelled (blue) samples. Data are derived from different fields of analysis and from replicate wafers (for details see Supplementary Table 1). Dashed lines represent literature values for the stable isotope natural abundance.

occupy ecological niches which would otherwise be uninhabitable to them. For assessing the dynamics and stability of microbial ecosystem functioning, fungi and bacteria need to be considered as interconnected entities rather than autonomously operating components of the ecosystem.

## Methods

**Microcosm setup.** Experiments were carried out in laboratory microcosms mimicking water and nutrient-rich ('with') and depleted ('without') zones (Fig. 1a). To create 'with' zones, two agar plugs (Ø 1 cm) were cut out from an FAB-medium agar plate (medium composition adapted from[41]) supplemented with 5 mM glucose as sole carbon source. These two plugs were placed in a petri dish (Ø 6 cm) at a distance of 1.4 cm from each other. Between the two plugs, a silicon wafer (Plano, Wetzlar, Germany) was placed representing the 'without' zone. This wafer served as a carrier for spores of *B. subtilis*, which were used as an indicator of changes in water and nutrient availability. Before its use, the wafer was cleaned with acetone, 70% ethanol and finally with autoclaved water and dried under a laminar flow box. Spore suspensions were bought from Merck (Darmstadt, Germany) and controlled microscopically for the absence of vegetative cells before use. Spores were washed three times with sterile-filtered, double-distilled water and in each washing step the supernatant was separated from the spores by centrifugation at 10,000*g* for 10 min. Subsequently, spores were resuspended in bidistilled water at a final concentration of $2.5 \cdot 10^6$ spores ml$^{-1}$ and 40 μl of the suspension were applied the wafer. Wafers were dried in an oven for 30 min at 70 °C.

**Quantification of total cells and spores.** The effects of mycelia in dry and nutrient-poor environments on the overall cell number and the abundance of spores were quantified using the fungi *F. oxysporum* or *Lyophyllum* sp. Karsten, or the oomycete *P. ultimum*. Therefore, a small amount of agar covered with mycelium from freshly overgrown FAB–glucose agar plates was cut out and transferred onto one of the agar plugs in the microcosms (Fig. 1a). Microcosms were sealed with parafilm, placed in a plastic container and incubated at 25 °C in the dark. The experiments were run in triplicates and control wafers were included. The control wafers were also inoculated with the spore suspension and placed within the microcosms but close to the edge of each petri dish to avoid contact with the mycelium. This should guarantee that all samples were exposed to the same water vapor pressure in the microcosm atmosphere. Mycelium-overgrown wafers and controls were excised from the microcosms after two (*P. ultimum*), four (*F. oxysporum*) and five (*Lyophyllum* sp. Karsten) days and transferred to glass tubes containing 0.5 ml 1 × PBS buffer (10 mM, pH 7). Cell detachment was carried out by vortexing the tubes for 1 min. Immediately, 400 μl were pipetted into new glass tubes and heated to 80 °C for 20 min to kill the vegetative cells[42]. Colony forming units in the heat-treated and untreated suspensions were counted using the drop plate method as described earlier[43]. Briefly, 10-fold dilution series of the supernatant were prepared directly in 96-well microtiter plates and five microlitres of consecutive dilutions were dropped on FAB-glucose agar plates containing 0.2% cycloheximid to suppress growth of the fungi and the oomycete. Plates were incubated at 25 °C for 48 h. Droplets giving rise to 5–30 single colonies were used to determine c.f.u. numbers.

**Visualization of spore germination in presence of mycelia.** Setups with *P. ultimum* were used to visualize germination of spores of *B. subtilis* in water and nutrient-free regions of the wafer. After the mycelium had overgrown the wafers, hyphae around the wafers were cut and the wafers were directly observed under the microscope (Axio Imager Z2, Carl Zeiss) using bright field illumination. Pictures were acquired with a Zeiss Axiocam 506 colour camera. To prove hyphae disintegration, samples were also stained with 4′,6-diamidine-2-phenylindol (DAPI)–amended mountant containing 9 parts Citifluor mountant (Citifluor, Leicester, United Kingdom), 1 part PBS (1 ×), and 1 μg ml$^{-1}$ DAPI. Epi-fluorescence images were acquired using DAPI filter settings.

**ToF–SIMS analysis.** In preparation for the envisaged nanoSIMS analysis a boron-doped silicon wafer overgrown with *P. ultimum* was analysed via ToF–SIMS technique employing a ToF-SIMS.5 (ION-TOF GmbH, Münster) instrument. The conditions of the ToF–SIMS experiment allowed for a detection of atomic and molecular secondary ions with *m/z* (mass to charge) ratio from 1 to about 500. The broad-range mass spectrum (1–500 amu) acquired for each raster point enabled us to compare the yield of different secondary ion species extracted from the sample. Detailed conditions of the analysis are specified as follows: the ToF–SIMS experiment has been performed using fast imaging mode of ToF-SIMS.5 operation in combination with delayed extraction[44,45] of negative secondary ions providing the Mass Resolving Power (MRP) above 3,000 and lateral resolution of about 130 nm. Under these experimental conditions, the 30 keV NanoProbe LMIG source was delivering 0.02 pA of primary $Bi_3^+$ cluster ions in 100 ns pulses with 200 μs repetition period. The analysis has been done in 400 scans (planes) with 5 shots of $Bi_3^+$ primary cluster ions per pixel distributed randomly in 512 × 512 raster over 56 × 56 μm sample area. The lateral distribution of ion yield was analysed using the proprietary ION-TOF SurfaceLab 6.5 software. The accumulation of acquired scans was done after lateral drift correction and the resulted total stack was analysed for the lateral distribution of the ion yield. The ToF–SIMS images produced were normalized by the total ion image.

**Analysis of water and nutrient transfer.** $^{13}$C-labelled glucose (99 atom%; Euriso-Top, Saarbrücken, Germany), $^{15}$N-labelled ammonium sulfate (99 atom%; Sigma, Munich, Germany) and $^{18}$O-labelled water (97 atom%; Campro Scientific, Berlin, Germany) were used for the labelling experiment. For the experiments with the labelled water, FAB-agar plates with 5 mM glucose were dried completely in an oven at 60 °C until only an agar foil was left. Thirty mg of this foil were transferred into an 5 ml Eppendorf tube and 2 ml of the labelled water were added. For the $^{13}$C and $^{15}$N labelling experiments FAB-agar plates without glucose and ammonium sulfate were prepared and again 30 mg were transferred to a 5 ml Eppendorff tube. Subsequently, 2 mg of $^{13}$C labelled glucose, 4 mg of $^{15}$N labelled ammonium sulfate and 2 ml autoclaved bidistilled water were added. The tube contents were incubated at 95 °C with continuous shaking at 700 r.p.m. until the agar was completely dissolved. Subsequently, two droplets of 400 μl were introduced into the petri dishes at a distance of 1.4 cm from each other and dried for 3 min under the laminar flow box until it formed hard agar blocks. The spores to serve as inoculum were prepared as described above. For the nanoSIMS experiment a conductive chromium-coated, round-shaped silicon wafer (10 mm diameter) was used as a spore carrier. Experiments were carried out using *P. ultimum* and inoculation was conducted as described above. Each labelling experiment was performed in duplicates. After 48 h, the wafers were removed from the microcosms and the biomass was fixed in an atmosphere over a solution of a 10% paraformaldehyde

and 37% ethanol at 30 °C for 2 h. Wafers were stored in a vacuum box until analysis. Parallel control experiments ($n = 2$) were performed with non-labelled compounds to assess the natural isotopic composition of the samples.

**NanoSIMS analysis.** The wafers were analysed with a NanoSIMS-50L instrument (CAMECA (AMETEK), Gennevilliers Cedex, France) to assess the transfer of the stable isotopes from the hyphae to the bacterial cells. Depending on the arrangement of hyphae, vegetative cells and spores, fields of analysis of $20 \times 20$, $30 \times 30$ and $40 \times 40\ \mu m$ were selected. Measurements were performed in negative extraction mode employing a DC source of primary $Cs^+$ ions. Detailed information on the analytical conditions for the measurements are described as follows: two replicate wafers labelled with $^{18}O$, and $^{13}C/^{15}N$ isotopes as well as two control wafers with natural isotopic composition were analysed. The 2 pA beam of $Cs^+$ ions was focused into $\sim 70\ nm$ spot at the sample surface during the analysis. The energy of $Cs^+$ collision with the sample was set to 16 keV in the analysis mode. The sample was scanned in a $512 \times 512$ px raster over analysis areas of $20 \times 20$, $30 \times 30$ or $40 \times 40\ \mu m$ with 5 msec dwell time per pixel. Before the analysis with 16 keV $Cs^+$ beam, the sample surface of $60 \times 60\ \mu m$ area was treated with 10 nA of low-energy (50 eV) $Cs^+$ beam for 10 min. The low-energy implantation/deposition of cesium has been performed with the purpose to equilibrate the working function for negative secondary ions and to make the outermost layer of the sample available for the analysis avoiding its sputtering during high-energy implantation. The secondary ions were analysed for their mass and charge ratio ($m/z$). Using the seven available collectors, the following secondary ion species were detected: $^{12}C^-$ (collector-1), $^{13}C^-$ (collector-2), $^{16}O^-$ (collector-3), $^{18}O^-$ (collector-4), $^{12}C^{15}N^-$ and $^{13}C^{14}N^-$ (collector-5), $^{32}S^-$ (collector-6), $^{31}P^{16}O_2^-$ (collector-7). Two secondary ion species ($^{12}C^{15}N^-$ and $^{13}C^{14}N^-$) were detected using the collector-5 by switching the deflector voltage in combined analysis mode. The mass resolving power (MRP) was checked to be between 8,000 and 12,000 ($M/\Delta M$) with the exit slit width of 40 $\mu m$, 20 $\mu m$ wide entrance slit, 200 $\mu m$ aperture slit and with the energy slit cutting 30% of secondary ions in high-energy tail of their energy distribution. It has been proven to get the sample sputtered completely within 35 scans upon the used analysis conditions; scans 2–30 were thus considered for the image and data analysis employing Look@NanoSIMS software (LANS)[46] allowing for stacking the scans with lateral drift correction and quantitative analysis of isotopic ratios ($^{18}O/^{16}O$, $^{13}C^{14}N/^{12}C^{14}N$ and $^{12}C^{15}N/^{12}C^{14}N$). Quantification of the atomic percent enrichment in $^{18}O$, $^{13}C$ and $^{15}N$ for hyphae, vegetative cells and spores was carried out by defining regions of interest around individual cells using the secondary electron images. The number of analysed fields and replicate wafers for each labelling experiment as well as the numbers of individual vegetative cells and spores used to quantify APE is specified in Supplementary Table 1.

**Data availability.** The data that support the findings of this study are available from the corresponding author upon reasonable request. The raw data sets for ToF- and nanoSIMS measurements have been deposited in the figshare repository under https://doi.org/10.6084/m9.figshare.4742857.v1 (url for ToF–SIMS data) and https://doi.org/10.6084/m9.figshare.4725409.v1 (url for nanoSIMS data), respectively.

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

## Acknowledgements

We acknowledge the Centre for Chemical Microscopy (ProVIS) at the Helmholtz Centre for Environmental Research supported by European Regional Development Funds (EFRE—Europe funds Saxony) for using their analytical facilities. This work was funded by the Helmholtz Association via the integrated project *Controlling Chemicals' Fate* of the research topic *Chemicals In The Environment* (CITE) within the research programme *Terrestrial environment*. The authors thank Prof J.D. van Elsas for kind provision of strain *Lyophyllum* sp. Karsten. We are grateful to Sophie Steigerwald, Manuel Trost, Rita Remer and Jana Reichenbach for skilled experimental help.

## Author contributions

All authors conceived the study framework and discussed the theoretical background. A.W., N.M., M.K. and L.Y.W. planned the study. A.W. and H.S. conducted the experiments and compiled the data. L.Y.W. and M.K. supervised the experiments. The manuscript was written by A.W. with extensive input from all authors.

## Additional information

**Competing interests:** The authors declare no competing financial interests.

