## [Peer Review File · Nature Communications]

Reviewers' Comments:

Reviewer #1 (Remarks to the Author):

This is a very interesting study providing intriguing insights into fungal-bacterial interactions. In a synthetic microcosm environment the authors showed (1) that the presence of fungal hyphae having access to water and nutrients promotes the germination of spores of *Bacillus* in an otherwise dry and nutrient-scarce environment and (2) for the first time the direct transfer of water and nutrients from fungal hyphae (of *Pythium ultimum*) to vegetative cells of *Bacillus subtilis* (based on nanoSIMS measurements of hyphae with bacterial cells attached).

The beauty of this study lies in its combination of quantitative methods and imaging techniques (ToF-SIMS, nanoSIMS), which allows not only to show that fungal hyphae promote bacterial germination, but also to explore the microscale spatial conditions and dynamics under which this is happening (only spores germinate that are less than 1.7 μm away from hyphae, bacterial vegetative cells closely attach to fungal hyphae). Although a nutrient and water transfer from fungal hyphae to bacterial cells has been assumed earlier (f.e. in soil) it has never been shown experimentally before.

The experiments are carefully designed, the results are clear, and the paper is written very well – I really enjoyed reading it. Although the study was based on a microcosm experiment in a controlled environment it has wider implications and gives food for thought: by demonstrating that fungal hyphae can act as powerful carriers helping bacteria to overcome drought and nutrient-limitations, it sheds new light on fungal-bacterial interactions in natural systems which will likely stimulate further research.

There are only a very few suggestions I would have to improve the manuscript:

L17 “thereby” makes no sense here, as there is no causality derived from the previous sentence.

Figure 4 and Figure 5: it would be helpful to indicate the natural abundance ratios of $^{18}\text{O}/^{16}\text{O}$, $^{13}\text{C}/^{12}\text{C}$ and $^{15}\text{N}/^{14}\text{N}$ in the respective scales (Fig 4b,d and Fig 5bc), or mention it (as number) in the figure legend.

Figure 5b: Unfortunately, the enrichment of the vegetative cells with ^{13}C is very difficult to observe from this picture because the green and the neighbouring blue colour look very similar. Is there a way to improve the colour contrast between these two enrichment levels? I am also wondering why is there a higher enrichment (light blue) of the background (wafer?) on the area left of the hyphae compared to the area beyond its right side? This is only the case for the ^{13}C , not ^{15}N .

Fig.6: To what does “field of analysis” correspond? Each to one $20 \times 20 \mu\text{m}$ area, as described in the supplementary methods? It would be helpful to also specify in the figure legend. In the supplementary method sections please also specify how many of these fields were analysed with nanoSIMS and from how many independent wafers (it just says “the samples”, but not how many). The same for the methods section of the paper (Analysis of water and nutrient transfer): not clear if replicate wafers were set up.

Supplementary Figure 1: Somethings wrong with the scale bar at the right end of the figure (double labelling)

Reviewer #2 (Remarks to the Author):

In summary, I recommend publication after minor revisions.

The interaction between bacterial and fungal communities represents a timely research topic in environmental sciences and in soil ecology. The paper by Worrich et al. shows that bacteria benefit from the transfer of water and nutrients in hyphae networks when bacteria are kept under low water and nutrient availability while the fungi have access to both water and nutrients. The transfer of water and nutrients in hyphal networks have been shown several times, but this paper - for the first time - shows, that bacteria in the vicinity of the hyphae can benefit from water and nutrients released from hyphae.

The authors use a laboratory approach with 3 different fungi and follow the response of the bacterium *B. subtilis* to fungal translocation and exudation. While the lab approach is artificial and the relevance of the findings for real natural conditions remains to be shown, the results are novel and of interest to a larger audience and especially to soil ecologists.

The paper is in general very well written and the methods used are at the forefront, as stable isotopic analyses of O, C and N at small scales (NanoSIMS) were used to directly demonstrate the uptake of nutrients and water and the triggered growth of the bacteria.

The experimental approach, data evaluation and presentation are very convincing to me and I cannot suggest major changes.

A few technical suggestions:

Fig1 c: The blue arrows can hardly be seen. Change blue to e.g. yellow

Fig. 3 can be omitted in my mind. It does not contribute significantly to the overall message and the patterns can hardly be seen.

Figure 6: These are data from NanoSims? Should be stated in the text. Moreover the legend at the right side of the graph for the different signatures is confusing. What is the meaning of the different signatures? Do you need the legend at all to show the effect? Can all green symbols be merged and compared to blue symbols?

Reviewer #3 (Remarks to the Author):

The manuscript aims to test the hypothesis that mycelia of fungi and oomycetes reduce water and nutrient stress for bacteria and enable bacterial activity in an otherwise dry environment. The authors use new techniques to study these processes at the level of a single cell (nanoSims).

They found the mycelia enabled both the germination of bacterial spores as well as vegetative growth near the hyphae. What is novel and very clever is that the authors used spatially resolved secondary ion mass spectrometry in combination with stable isotope labeling to show that there was transport of labeled compounds in mycelia and a supply of water, carbon and nitrogen to the cells of *B. subtilis* located close to the hyphae.

The techniques used here are cutting edge and the outcome effectively demonstrates the power of these methods. However, how novel is this interaction between hyphae and *B. subtilis*? We know that *B. subtilis* colonizes plant roots in a similar manner. The fact that all 3 types of mycelium were implicated, highlights the generality of the phenomenon. Therefore, I was left wondering: is there something special about the fungi, or is it a demonstration of leakiness of any living biological matter, more broadly? Would it be useful if the authors tested as a control, other types of biological matter? Could they use artificial networks that leaked nutrients and water. Would they find something similar? For example, recent work by Worrich et al. (2016) found positive effects on artificial mycelium-like dispersal networks on bacterial dispersal and growth.

Also, were there any attempts to study how the diffusion of nutrients and water affected the growth/fitness of the mycelium itself? Is this commensalism or parasitism? Passive or active?

My specific comments are below:

The first paragraph could be better written. It contains many ideas and no clear structure of thought. This should establish the background and the ideas to be tested, what is known versus what is unknown.

Line 40: environment needs an 's'

Line 43: Is the encouraged format to summarize the results in the intro before the results section?

Line 56: "to what degree"

Line 99: Are these biomass analyses critical? It seems they are important to make a convincing argument about transfer processes. The key is to demonstrate that the movement of nutrients was important/significant given a certain amount of biomass.

Line 111: Did isotopic replacement modify the growth or change the dynamics of *P.ultimum*? Could this have changed the leakiness of the mycelium to the bacteria. More details here are necessary.

Line 154: Here a summary (i-iii) of three points were made. However point (ii) in particular seems weak. The authors need to explain how the micro-scale spatial organization of hyphae and governed the outcome of the activation. That wasn't clear from the results section. Also, is this result that surprising given that this spatial organization is what is usually observed for the interaction of *B. subtilis* with plant roots.

Line 159: The authors argue that the study provides for the first time direct experimental evidence for stimulation of bacterial activity by mycelium. But they also point to a wealth of literature, including the paper by Warmink that demonstrated an increased number of culturable bacteria in the vicinity of fungal hyphae in soil microcosms.

Line 179: Similarly, because a single paper did not include bacterial-fungal interactions is not evidence that "fungal-bacterial interactions have most often been disregarded". At brief look at the literature shows a huge variety of studies demonstrating positive interactions among fungi and bacteria. It is important not to oversell the novelty of the findings.

Line 182: New paragraph is needed

Line 205: the authors use the phrase "exchanging organisms" – but do they show that the bacteria are exchanging anything for the water and nutrients? Instead, they seem to demonstrate a diffusion from the fungi into the bacteria which are located in close proximity to the mycelium.

Line 221: This is similar to the concluding line, where "exchange" is used again. Isn't diffusion more appropriate? Also exchange implies active, where diffusion is passive.

Methods: Was the differential growth of the mycelium taken into account? Meaning, how did the author standardize for amount of mycelium growing from the plug? Was this quantified?

Figures are elegant and easy to read. Methods was perfect level of detail.

Point-by-point answers to reviewers comments

Reviewer 1:

L17 “thereby” makes no sense here, as there is no causality derived from the previous sentence.

We considered the reviewer comment and also in accordance to the suggestion made by reviewer 3 we rewrote the first paragraph of the introduction (cf. ll. 14 – 27)

Figure 4 and Figure 5: it would be helpful to indicate the natural abundance ratios of $^{18}\text{O}/^{16}\text{O}$, $^{13}\text{C}^{14}\text{N}/^{12}\text{C}^{14}\text{N}$ and $^{12}\text{C}^{15}\text{N}/^{12}\text{C}^{14}\text{N}$ in the respective scales (Fig 4b,d and Fig 5bc), or mention it (as number) in the figure legend.

We are grateful to the reviewer for this suggestion. We changed Fig. 4 and 5 and their captions accordingly by including the natural abundance in the color bars.

Figure 5b: Unfortunately, the enrichment of the vegetative cells with ^{13}C is very difficult to observe from this picture because the green and the neighboring blue color look very similar. Is there a way to improve the color contrast between these two enrichment levels?

We appreciate the reviewer comment and considered the suggestion. The color gradation was already chosen to obtain the highest contrast between hyphae, vegetative cells and background signal. In order to improve visibility, we outlined vegetative cells and spores by black and white circles, using the secondary electron image as template. We think that this makes it easier to distinguish between hyphae, neighboring cells and background signals in Fig. 5b and c.

I am also wondering why is there a higher enrichment (light blue) of the background (wafer?) on the area left of the hyphae compared to the area beyond its right side? This is only the case for the ^{13}C , not ^{15}N .

*During sputtering, only a small part of about 1% of the sample material is ionized and available for the extraction to the mass spectrometer. The sputtered neutral particles are, however, redeposited over the analysis area. The re-deposition pattern depends on the geometry of the sample and thus can take place preferably on one site. In addition, the localization of *Bacillus subtilis* cells (less enriched in ^{13}C) on the right side may also obscure re-deposition. The re-deposition is more visible in the ^{13}C image because i) the higher ^{13}C enrichment of the hyphae in comparison with the ^{15}N enrichment ii) the different scaling for ^{13}C and ^{15}N (0.6 vs 1).*

We also added a short explanation in the respective figure legend (cf. ll. 368-370):

“The higher background signal for ^{13}C on the left side of the hyphae (b) results from the topography-dependent re-deposition of sputtered sample material”.

Fig.6: To what does “field of analysis” correspond? Each to one 20x20 μm area, as described in the supplementary methods? It would be helpful to also specify in the figure legend.

We thank the reviewer for this remark. Field of analysis corresponds to each individual area that was analyzed by NanoSIMS. Depending on arrangement of hyphae cells and spores, we adapted the size of the raster to either 20 x 20, 30 x 30 or 40 x 40 μm . We specified the size of the analyzed fields in the Materials and Methods part of the manuscript as well as in the respective figure legends.

ll. 310 – 311: “Depending on the arrangement of hyphae, vegetative cells and spores, fields of 20 x 20, 30 x 30 and 40 x 40 μm were selected.”

In the supplementary method sections please also specify how many of these fields were analyzed with nanoSIMS and from how many independent wafers (it just says “the samples”, but not how many). The same for the methods section of the paper (Analysis of water and nutrient transfer): not clear if replicate wafers were set up.

We agree to this point and decided to include a table in the Supplementary information specifying the number of analyzed fields and wafers and also how many spores and vegetative cells were used to calculate APE. This helped us also to address the comment of reviewer 2 and to simplify the legend in Fig. 6 by distinguishing only between labeled and non-labeled samples and show all the additional information in the SI. In the methods section we included the information that duplicate wafers were set up for each treatment and the controls, respectively.

l.302 - 303: Each labeling experiment was performed in duplicates.

ll.305 - 306: Parallel control experiments (n=2) were performed with non-labeled compounds to assess the natural isotopic composition of the samples.

Supplementary Figure 1: Somethings wrong with the scale bar at the right end of the figure (double labelling)

We thank to the reviewer for pointing this out. The line numbers moved to the right end of the page during conversion to pdf. We fixed this in the new version of the manuscript.

Reviewer 2:

Fig1 c: The blue arrows can hardly be seen. Change blue to e.g. yellow

We agree to this remark and changed the color to yellow.

Fig. 3 can be omitted in my mind. It does not contribute significantly to the overall message and the patterns can hardly be seen.

We also discussed this issue in advance to preparing the manuscript, however, ToF-SIMS was applied to map the elemental composition of the sample and this was a crucial step in preparation for the NanoSIMS analyses. Based on these measurements we were able to identify suitable ion species to track resource transfer processes. It became obvious that O- and CN- represent suitable candidates to visualize O and C transfer, while PO₂⁻ and CH are rather inappropriate. Although images are not that clear as in the NanoSIMS analyses, we argue that it is already an experimental asset and a huge technical advancement to visualize and distinguish individual bacterial cells, spores and hyphae with ToF-SIMS, which is usually restricted in its application in microbiology due to its drawbacks of obtaining both high mass resolution and high spatial resolution with adequate sensitivity. Therefore, we think that it is worth to keep figure 3 in the manuscript as it makes the paper also attractive to readers interested in the spatially-resolved mass spectrometric analysis of microbes using combinations of ToF- and NanoSIMS.

Figure 6: These are data from NanoSims? Should be stated in the text. Moreover the legend at the right side of the graph for the different signatures is confusing. What is the meaning of the different signatures? Do you need the legend at all to show the effect? Can all green symbols be merged and compared to blue symbols?

We thank the reviewer for this remark. The data were obtained from NanoSIMS measurements, which is now indicated in the figure legend. We also simplified Fig. 6 according to the suggestions. The information on the number of analyzed fields, replicate wafers, numbers of spores and vegetative cells are now included in Supplementary Table1. We hope that this improved the readability of the figure considerably.

ll. 381 – 384: Atom Percent Enrichment (APE) for (a) ¹⁸O, (b) ¹³C and (c) ¹⁵N by single cells and spores of B. subtilis measured with NanoSIMS in labeled (green) and non-labeled (blue) samples.

Reviewer 3:

However, how novel is this interaction between hyphae and B. subtilis? We know that B. subtilis colonizes plant roots in a similar manner.

The aim of our study was to evaluate whether mycelia can overcome bacterial water and nutrient stress in a dry and oligotrophic microhabitat. Towards this goal we were able to provide clear visual and quantitative evidence (using different stable isotope tracer compounds and ToF- plus nanoSIMS approaches) for multiple transfer of water and nutrients from various fungal mycelia towards initially non-interacting biota exposed to non-favorable conditions. To the best of our knowledge, both the research question (that is highly relevant for microbial ecosystem functioning) and the experimental approaches used are unprecedented. Bacillus subtilis spores

were used as model biota as they are commonly found in terrestrial habitats (e.g. in the rhizosphere) as commented by reviewer 3.

The fact that all 3 types of mycelium were implicated, highlights the generality of the phenomenon. Therefore, I was left wondering: is there something special about the fungi, or is it a demonstration of leakiness of any living biological matter, more broadly?

We agree with the comment of reviewer 3 that the observed transfer of water and nutrients across cell walls of clearly differing mycelial organisms points at a wide-spread phenomenon. However, these transfers may vary for different fungi and that is why we tested different species. In addition, these findings have to be seen against the background that (mycelial) fungi are abundant and quasi everywhere in terrestrial habitats. Fungi embody up to 75% of the subsurface microbial biomass and hyphae of fungi create dense fractal networks of up to 10^4 m length per g of topsoil. Their mycelia further penetrate micro-aggregates by wedge-shaped hyphae and allow maximal pervasion and mobilization of resources. It is hence not surprising that fungal lifestyles also often match situations found in extreme habitats, such as in desiccation, hydrostatic pressure or extreme pH. These examples show that mycelial fungi have a unique lifestyle and occupy specific ecological niches at extents and spatial densities not observed in other organisms. The idea of reviewer 3 that also other living organisms may 'leak' (e.g. the exchange of water between associated bacteria) is interesting but beyond the scope of our research.

Would it be useful if the authors tested as a control, other types of biological matter?

The emphasis of our study was to demonstrate that mycelia may reduce bacterial water and nutrient stress in dry and oligotrophic microhabitats. Mycelial fungi exhibit a unique lifestyle and are cornerstones of soil ecosystem functioning. Focusing on mycelia hence is justified. As outlined above, in depth analysis of water and metabolite exchange among other organisms is certainly interesting but clearly beyond the scope of our study.

Could they use artificial networks that leaked nutrients and water. Would they find something similar?

*The novelty of our study is to show that hyphae contribute to the stress reduction of bacteria exposed to dry and oligotrophic microhabitats. *Bacillus subtilis* spores were used as initially inactive model biota as they are known to germinate in presence of sufficient provision of water and nutrients. Using artificial networks (even if they existed and provided the density and complexity of mycelia) which 'leak' water and nutrients may hence not add significant novelty to already existing knowledge. In our opinion mycelia are in order to account for both physical and (eco-)physiological processes (e.g. the dynamic behavior of mycelia in response to changing environmental conditions or the bi-directionality of the cytoplasmic streaming).*

For example, recent work by Worrich et al. (2016) found positive effects on artificial mycelium-like dispersal networks on bacterial dispersal and growth.

The study by Worrich et al. (2016) mentioned by reviewer 3 analyzed the effects of (inert) model networks on the dispersal and the biodegradation performance of bacterial cells at differing water potentials. The networks tested hence did not facilitate any internal transport or leaking of resources as shown in the present study and we do not know any system being able to do so.

Also, were there any attempts to study how the diffusion of nutrients and water affected the growth/fitness of the mycelium itself? Is this commensalism or parasitism? Passive or active?

As the B. subtilis spores are initially metabolically inactive, they thus could not foster the release of resources from the mycelium. The initial transfer of resources to the bacteria is very unlikely to exert any fitness diminishing effects on the mycelium and would occur similarly in absence of bacteria. After germination, however, vegetative cells of B. subtilis start to interact with the mycelium to further increase the release of compounds from the mycelium. Thus, for vegetative cells, active and passive interactions are worth considering, because bacilli could alter fungal membrane permeability to increase or modify nutrient efflux (de Boer et al., 2005). Indeed, this might reduce the fitness of the fungus/oomycete. However, we observed vegetative cells also along intact hyphae (cf. Fig 1), which indicates that the interaction is not solely antagonistic. The different possibilities by which bacteria could obtain nutrition from the fungus were discussed in the text and hypothesis on their occurrence in the experiment were drawn (cf. ll. 174 - 196). However, the explicit type of interaction was not tested as it will largely depend on the chosen organisms. This was not the focus of the present study. The mechanism shown here is applicable to metabolically inactive bacteria, which cannot provoke nutrient release by fungi but rather experience a beneficial change in their local environmental conditions due to mycelia-based resource provisioning.

The first paragraph could be better written. It contains many ideas and no clear structure of thought. This should establish the background and the ideas to be tested, what is known versus what is unknown.

We thank the reviewer for this suggestion. We rewrote the first paragraph and omitted redundant and unnecessary information. We further tried to focus more on the current knowledge gaps.

ll. 24 – 27: While recent data suggest that mycelia create hospitable microhabitats for bacteria due to the exudation of carbonaceous compounds and a moistening of the surrounding substrate, experimental evidence for a stimulation of bacterial activity in harsh environments is lacking.

See also highlighted changes version of the resubmission.

Line 40: environment needs an ‘s’

Corrected as requested.

Line 43: Is the encouraged format to summarize the results in the intro before the results section?

Yes, it is one of the points in the checklist for Nature Communications articles.

Line 56: “to what degree”

Corrected as requested.

Line 99: Are these biomass analyses critical? It seems they are important to make a convincing argument about transfer processes. The key is to demonstrate that the movement of nutrients was important/significant given a certain amount of biomass.

The biomass analyses were important to obtain information on the elemental sample composition and to identify suitable secondary ion species for the analysis of the transfer processes. As indicated in the response to reviewer 2, we think that this is also a methodological advance for the analysis of bacteria with ToF-SIMS.

Line 111: Did isotopic replacement modify the growth or change the dynamics of *P. ultimum*? Could this have changed the leakiness of the mycelium to the bacteria. More details here are necessary.

*We observed that *P. ultimum* was growing slower when provided with all three labelled compounds simultaneously. We interpreted this as a cumulative, strong kinetic isotope effect: many enzymatic reactions proceed faster with non-labelled substrates than with substrates enriched in heavy isotopes, which is the basis of stable isotope fractionation (e.g. Meckenstock et al., 2004; Elsner et al., 2005). However, leakiness of the mycelium cannot be affected. In order to minimize any kinetic isotope effects on enzymes, we performed parallel labelling experiments.*

*We modified the main text to read: “*P. ultimum* was grown in parallel experiments with ¹⁸O-labelled water or with a combination of ¹³C-glucose and ¹⁵N-ammonium sulfate, to prevent slow development of hyphae due to strong kinetic isotope effects (cf. ll. 106 – 109).”*

22. Meckenstock RU, Morasch B, Griebler C, Richnow HH. Stable isotope fractionation analysis as a tool to monitor biodegradation in contaminated aquifers. *Journal of Contaminant Hydrology* 75, 215-255 (2004).

23. Elsner M, Zwank L, Hunkeler D, Schwarzenbach RP. A new concept linking observable stable isotope fractionation to transformation pathways of organic pollutants. *Environmental science & technology* 39, 6896-6916 (2005).

Line 154: Here a summary (i-iii) of three points were made. However point (ii) in particular seems weak. The authors need to explain how the micro-scale spatial organization of hyphae and

governed the outcome of the activation. That wasn't clear from the results section. Also, is this result that surprising given that this spatial organization is what is usually observed for the interaction of *B. subtilis* with plant roots.

The points (i – iii) represent a short summary of the major results. We agree that point ii) was not phrased very precisely. We therefore rephrased this point to exactly what was observed and stated in the results:

l. 154“[...] ii) only spores in close vicinity to the hyphae could germinate [...].”

Later in the discussion we refer to each point again and present some explanations. We also refer again to the micro-scale spatial distribution by explaining that close proximity between fungi and bacteria is needed for the transfer of resources (cf. ll. 197 - 200). It is true that this is observed for plant roots although this does not automatically imply that it occurs with fungi as well.

Line 159: The authors argue that the study provides for the first time direct experimental evidence for stimulation of bacterial activity by mycelium. But they also point to a wealth of literature, including the paper by Warmink that demonstrated an increased number of culturable bacteria in the vicinity of fungal hyphae in soil microcosms.

There are indeed different studies reporting an increased abundance of bacteria in the mycosphere and it is hypothesized that fungal exudates favor the growth of bacteria. However, multiple resource transfer of water and nutrients has not been directly experimentally demonstrated (see also initial comment to reviewer 3). Here we could show for the first time that water and nutrients transported in the mycelium over distances of centimeters, are subsequently transferred to bacteria, which is sufficient to regain full activity by the supply of scarce resource. Therefore, we argue against the change of the sentence as we clearly stated that it is the direct evidence of the supply of scarce resources in a completely dry and oligotrophic environment which makes the study novel.

Line 179: Similarly, because a single paper did not include bacterial-fungal interactions is not evidence that “fungal-bacterial interactions have most often been disregarded”. At brief look at the literature shows a huge variety of studies demonstrating positive interactions among fungi and bacteria. It is important not to oversell the novelty of the findings.

We agree with the reviewer and apologize for this overly simplified statement. We removed this paragraph from the manuscript.

Line 182: New paragraph is needed

A new paragraph was inserted (cf. l. 174).

Line 205: the authors use the phrase “exchanging organisms” – but do they show that the bacteria are exchanging anything for the water and nutrients? Instead, they seem to demonstrate a diffusion from the fungi into the bacteria which are located in close proximity to the mycelium.

We thank the reviewer for this remark and agree that “exchanging” implies that bacteria also give something to the fungus. We rephrased the sentence to:

“Thus, close proximity between the mycelium and the bacterial cells organisms constituted a prerequisite for diffusion-based transfer of resources as observed in different studies on nutrient exchange or chemical communication in microbial communities.”(cf. ll. 197 - 200)

Line 221: This is similar to the concluding line, where “exchange” is used again. Isn't diffusion more appropriate? Also exchange implies active, where diffusion is passive.

We also rephrased this sentence using “transfer” instead of “exchange”:

ll. 215 – 217: “Although the combination of the selected organisms may be artificial, we observed similar net effects for all three mycelial organisms likely indicating widespread nutrient and water transfer from mycelia to bacterial cells.”

Methods: Was the differential growth of the mycelium taken into account? Meaning, how did the author standardize for amount of mycelium growing from the plug? Was this quantified?

The differential growth was not taken into account and the amount of mycelia covering the waver was not standardized. Although detailed quantitative analysis of the effectiveness of differing mycelia and their abundance on spore germination may be interesting, it was not central to our study. We focused on sufficient supply for germination and our experiment was designed to serve as a proof-of-principle study to show the effects of mycelia of clearly differing organisms on spore germination.

*We agree to the reviewer comment that both the cell wall composition and mycelial growth may explain the differences observed between the mycelia tested. Hence, we changed the discussion section as follows: ll. 212 – 215: “**The differences observed for total CFU and the amount of germinated cells may be explained by the differences in the mycelial coverage of the waver or differing cell wall compositions of oomycetes and fungi such as the lack of chitin in *P. ultimum*.**”*

Reviewers' Comments:

Reviewer #3 (Remarks to the Author):

I am satisfied with the author's edits and responses.